# Health-Related Expectations Regarding Aging among Middle-Aged and Older Japanese: Psychometric Performance and Novel Findings from the ERA-12-J

**DOI:** 10.3390/ijerph192013509

**Published:** 2022-10-19

**Authors:** Michael Annear, Yasuo Shimizu, Tetsuhiro Kidokoro

**Affiliations:** 1Faculty of Sport Sciences, Waseda University, Nishitokyo 202-0021, Japan; 2Division of Arts and Sciences, International Christian University, Mitaka 181-8585, Japan; 3Research Institute of Physical Education, Nippon Sport Science University, Setagaya-ku 158-8508, Japan

**Keywords:** health expectancy, healthy aging, physical health, mental wellbeing, cognitive function, Asia, Japan

## Abstract

Background and objectives: Health-related expectations regarding aging is a gerontological construct that is potentially predictive of morbidity and mortality in later life. The Expectations Regarding Ageing scale (ERA-12) is a widely used measure of health-related expectations, although it has not previously been administered in Japanese. The present research aimed to elucidate the psychometric properties of the first Japanese translation of the ERA-12 and evaluate health-related expectations among middle-aged and older Japanese. Research design and methods: Repeated online surveys were conducted with representative quota samples of middle-aged and older adults in Tokyo during 2021 (*N* = 1600). Primary outcome measures included total and subscale scores on a Japanese translation of the ERA-12 (ERA-12-J) addressing perceptions of physical, mental, and cognitive health. Standard measures were also used to gather information regarding respondent demographic details, general health, and health-related behavior. Results: The ERA-12-J and associated subscales showed acceptable test-retest reliability (*t*(1598) = 0.60, *p* = 0.63), internal consistency (α > 0.80), inter-item correlation (*r* = 0.21–0.78) and item-total correlation (*r* = 0.53–0.73). Confirmatory Factor Analysis verified the hypothesized three-factor structure and construct validity on four common indices of fit (GFI = 0.968; CFI = 0.978; AGFI = 0.950; RMSEA = 0.059). ERA-12-J scores among Japanese respondents revealed prevailing negative sentiments concerning physical and cognitive health, with less negative sentiment regarding mental health. Significant and independent differences emerged concerning gender and age cohort, with middle-aged adults and females holding more negative expectations about their future health. Discussion and implications: The ERA-12-J provides a sound basis for the elucidation of health-related expectations about aging in Japan and a useful tool for international comparative studies. Education and workplace intervention may be required in Japan to address age and gender disparities in health-related expectations.

## 1. Introduction

### 1.1. Population Aging and Health in Japan

Japan leads the world in population aging and has been described as a super-aged society in reference to the rising proportion of the population aged 65 or older [1]. This cohort comprises more than 28% of the current population and is projected to exceed 40% by the middle of the century [2]. While population aging reflects Japan’s successes overcoming childhood diseases and managing chronic morbidity, it also brings challenges for health and social care systems, which face increasing demand and dependency in concert with declining workforce participation [1]. The COVID-19 pandemic has highlighted the vulnerability of aging societies and called into question the prevalence of institutional care and the biomedical model of treatment for age-related diseases, which prioritizes medical and pharmaceutical intervention over lifestyle change [3]. More understanding is required about the psycho-social determinants of health and health-related behaviors, which have potential to compress morbidity in later life and foster greater participation across the life course. Such research is particularly important in Japan, where many adults live long lives, but often in poor health. Epidemiological evidence gathered over three decades in Japan has shown that life expectancy gains are outpacing health expectancy among older cohorts leading to an expansion of morbidity [4,5]. There are also reported age and gender discrepancies in health behaviors and outcomes that reflect complex cultures of male-dominance and overwork that pervade modern Japan [6,7].

### 1.2. Health-Related Expectations

Health-related expectations regarding aging is a gerontological construct that addresses an individual’s self-referential and health-related perceptions about their future physical capacity, mental wellbeing, and cognitive function [8]. While there is a growing body of literature concerning the more general concept of attitudes about aging (including broad psychosocial components such as fear of older adults, physical appearance changes, received respect, stereotypes, and agism) [9,10,11], expectations regarding aging is a comparatively unique concept due to its focus on specific components of health [12]. Expectations regarding aging are hypothesized to affect health behaviors via an individual’s beliefs (social norms and stereotypes) [13], knowledge (understanding about the aging process) [14], and sense of self-efficacy (perceptions of individual capabilities) [15]. Health-related expectations regarding aging are measured using the Expectations Regarding aging scale (ERA-12)—a valid and reliable tool developed in the United States [16]. 

### 1.3. Research on Expectations

The ERA-12 has been validated in English, Chinese, Korean, Turkish, Persian, Tamil, and Malay [16,17,18,19,20,21]. This measure has been primarily applied within preventive medicine and clinical gerontology to understand precursors of health behaviors and to inform clinical service delivery improvements for older adults [14,17,22]. Studies of middle-aged and older adults undertaken across multiple countries using the ERA-12 have identified prevalent negative expectations about aging on physical, mental, and cognitive function subscales [17,19,21]. It appears, however, that health-related expectations are influenced to some extent by individuals’ adherence or avoidance of health behaviors. For example, research undertaken with English-speaking samples of older adults has identified that more positive expectations regarding aging are associated with an increased likelihood of receiving a physician examination within the past two years [15]. Other research has reported that more positive expectations regarding aging appear to be associated with self-reported physical activity and functional ability [23,24,25]. Internationally, associations have also been observed between ERA-12 scores and age, gender, ethnicity, and self-rated health [16,22,26,27]. 

### 1.4. Research Objectives

Despite sustained population aging and the highest average lifespan among developed nations, there has been no examination of health-related expectations regarding aging among middle-aged and older Japanese using reliable and valid measures. This is important as such expectations may be an important precursor of health behavior change and potentially predictive of morbidity. Consequently, the present study had two research objectives: (1) evaluate the psychometric properties of the first Japanese translation of the Expectations Regarding Aging scale; and (2) elucidate expectations among middle aged and older women and men in Tokyo across physical, mental, and cognitive health domains.

## 2. Materials and Methods

### 2.1. Design and Conceptual Framework

The research was informed by repeated online surveys conducted during 2021 with quota samples of middle-aged and older adults in the greater Tokyo region. The conceptual model for the measure was based on the existing 12-item English-language scale (comprising three health-related subscales and an overall summative measure). The development of the ERA-12 measure was underpinned by hypothesized and observed relationships between older adults’ perceptions of aging and their future health outcomes [28,29]. Written approval for the use the ERA-12 scale was provided by the author of the original development paper [16]. 

### 2.2. Sampling and Selection

A population of 146,000 middle-aged and older adults in greater Tokyo were registered users of a research subsidiary (Rakuten Insight) of the largest online eCommerce platform in Japan (Rakuten Inc., Tokyo, Japan) and had consented to selection for research studies in 2021. An email invitation was initially sent from the survey vendor to all eligible respondents (middle aged and older men and women in Tokyo who were registered for research participation). Respondents who volunteered to complete the survey clicked on a URL link in the invitation email, read an information page about the study, and then completed and submitted the online survey. The first 800 eligible respondents to reply to the survey who meet quota requirements for age and gender were included in the study. A further data integrity criterion was also applied during the data collection phases to exclude surveys where completion times were >30% faster than the median completion time. This exclusion criterion is used to identify and remove any responses that were potentially erroneous. Minimum sample size was determined using G*Power software [30] and was undertaken to exceed requirements for achieving a 95% confidence level (*N* ≥ 400). An independent, follow-up sample was generated (using exactly the same quota sampling approach as described above) with a new cohort of respondents in October 2021 to facilitate test-retest analysis of the translated measure and to conform to ethical obligations for respondent anonymity and confidentiality. 

Survey administration dates in March and October were selected for two practical reasons. Firstly, both data collection times occurred in the weeks following the end of official government state of emergency periods related to the COVID-19 pandemic. Therefore, there were no restrictions on personal travel or activities during the survey period, which may have otherwise influenced reporting or behavior among the target sample. Secondly, conducting research during March and October avoided any extremes in weather and climate (e.g., rainy season or heatwaves), which may have influenced responses related to health or behavior. The data collection period ran for approximately two weeks at both time points. All respondents read an online information page prior to study commencement and were informed that submission of the survey constituted consent for de-identified and aggregated data to be used in research. Study procedures were reviewed and approved by an institutional human research ethics committee prior to the administration of both surveys (reference 2020-344/2021-235).

### 2.3. Measures

The primary outcome measure, the ERA-12 scale, has demonstrated acceptable reliability (α = 0.89) and validity among cohorts of middle-aged and older adults [16]. The scale consists of 12 statements about aging expectations divided into three subscales: physical health, mental wellbeing, and cognitive function. All scale statements express a negative sentiment about aging (e.g., *I expect that as I get older I will become more forgetful*) scored on a four-point Likert-type scale (1 = definitely true, 2 = somewhat true, 3 = somewhat false, 4 = definitely false). Total and subscale values are expressed as standardized scores ranging from 0–100, where lower scores indicate more negative expectations. Self-rated health was evaluated using the BEHS-6 measure of holistic health, which is a short-form (6-item) Japanese-language measure based on the World Health Organization’s enduring definition (α = 0.92) [6]. Self-reported daily sitting time was used as a proxy for health behavior and was assessed with a single-item question from the valid and reliable translation of the International Physical Activity Questionnaire [31]. In reference to ongoing COVID-19 transmission, a single-item Likert-type question was also used to assess respondents’ beliefs about the extent to which their health had been affected by the pandemic. Finally, standard demographic questions gathered descriptive data, including age, gender, education level, employment status, living circumstances, and nationality. 

### 2.4. Translation and Back Translation

The ERA-12-J was developed by a native English speaker (M.A.) and two bilingual Japanese speakers (Y.S. and T.K.) based on the original English-language version [16]. Following an initial translation of items from English to Japanese an iterative process of back translation was undertaken. The back translation process involved a discussion between all authors whereby Y.S. and T.K. (bilingual Japanese and English speakers) explained the meaning of each Japanese item in English while M.A. (a native English speaker) confirmed that this description matched closely with the original version to ensure no loss of item meaning [32]. 

### 2.5. Analysis

Following preliminary data cleaning (checks for incomplete surveys, erroneous responses, or completion times >30% faster than the median response time), distribution of the main outcome variable (overall ERA-12-J score) was explored to confirm normality. Descriptive statistics were used to present demographic information about the sample. The process for assessing the psychometric performance of the ERA-12-J consisted of three procedures that aligned with the development of the English-language measure (Sarkisian et al., 2005) [16]. These included measures of internal consistency reliability (Cronbach’s alpha coefficient), test-retest reliability, and construct validity. Independent samples *t*-tests were used to assess test-retest reliability and correlation coefficients were used to evaluate inter-item and item-total relationships. Confirmatory factor analyses (CFA) were performed to evaluate whether the three-subscale model hypothesized by Sarkisian et al. [16] fit the data from a Japanese sample. A series of two-way between groups ANOVA were also conducted to explore potential differences and interactions between age and gender in relation to ERA-12-J scores (known groups evaluation). Correlations were also assessed between ERA-12-J scores and three health-related scales to explore the alignment with related health and behavioral variables. All analyses were performed in IBM SPSS (version 27, IBM Corp., Armonk, NY, USA).

## 3. Results

The respondent sample was drawn from a population of registered users of the research subsidiary of Japan’s largest ecommerce platform. At the time of the study within Tokyo there were 102,000 registered research ‘monitors’ aged 45–64 (*F* = 45,000; *M* = 57,000) and 44,000 aged 65 and older (*F* = 16,000; *M* = 28,000). From this sampling pool, a total quota sample of 1600 online respondents from greater Tokyo was attained over two time periods in 2021. Following initial data evaluation, the distribution of ERA-12-J scores conformed to expectations of normality and outlier influence was determined to be minimal. Respondent populations were comparable across the two data collection phases, and independent samples *t*-test results for mean age revealed no significant difference at times 1 and 2, *t*_(1598)_ = 0.87, *p* = 0.39. Demographic characteristics for both samples are summarized below in Table 1. 

### 3.1. Internal Consistency Reliability

An independent-samples *t*-test was conducted to compare overall score on the ERA-12-J undertaken at two different times during 2021 (six-month interval) to assess test-retest reliability (see Table 2 below). There was no significant difference in the mean ERA-12-J total score between time one (*M* = 36.80, *SD* = 15.13) and time two (*M* = 37.16, *SD* = 15.18; *t* (1598) = 0.60, *p* = 0.63, two-tailed). Internal consistency reliability was also assessed for the overall and subscale scores at time 1 and time 2 and scores were found to be consistently high (α > 0.80). Inter-item correlations ranged from 0.21 and 0.78 indicating positive item association without redundancy. Item-total correlations also showed good alignment with values ranging from 0.53 to 0.73 across all items at two time periods.

### 3.2. Construct and Factorial Validity

To determine whether the ERA-12-J accurately reflected the three-factor structure of the original ERA-12 [16], a CFA was performed using Structural Equation Modelling (see Table 3 below). Estimated intercorrelation between latent factors was satisfactory with all below the acceptability criterion of 0.80 [33]. All beta coefficients were significant (*p* < 0.001) and ranged from 0.61–0.89. The 12-item ERA-12-J exhibited good model fit across four common indices (GFI = 0.968; CFI = 0.978; AGFI = 0.950; RMSEA = 0.059). The chi-square for the model was significant (χ^2^ = 324.300, *df* = 50, *p* < 0.001), although this was expected and is considered an unreliable indicator of fit in the present analysis due to the larger sample size (*N* = 1600) [34,35]. 

### 3.3. ERA-12-J Total and Subscale Scores

The mean total ERA-12 score was 36.98 out of 100 (*SD* = 15.15). Mean subscale scores were as follows: physical health = 28.31 (SD = 16.63), mental health = 48.03 (*SD* = 20.03), and cognitive health = 34.60 (*SD* = 17.56), indicating prevailing negative perceptions of aging processes. A series of two-way between groups ANOVA were conducted to evaluate the relationship and potential interaction effects between age, gender, and ERA-12-J total and subscales scores and as a check of known groups validity. With regards to the total score on the ERA-12-J, the interaction between age and gender was not significant, *F*(1, 1596) = 0.058, *p* = 0.81. There was a statistically significant main effect for age with older adults exhibiting a higher score (less negative) compared to middle-aged adults, *F*(1, 1596) = 11.61, *p* < 0.001, with a small effect size (partial eta squared = 0.01). There was also a statistically significant main effect for gender whereby males had a significantly higher ERA-12-J total score, *F*(1, 1596) = 14.54, *p* < 0.001, with a small effect size (partial eta squared = 0.01). Subscale values were mostly in line with results observed for the overall score and are summarized below in Table 4. The only exception was the non-significant difference in cognition scores between middle-aged and older adults. 

### 3.4. Potential Associations with Self-Rated Health and Health-Risk Behavior

Relationships were also explored between ERA-12-J scores and three health-related variables: self-reported general health, self-reported daily sitting minutes, and self-reported health impact of the COVID-19 pandemic. This analysis was undertaken as an additional check of construct validity to assess whether health-related expectations were associated with other health-related measures as has been reported internationally. ERA-12-J total scores correlated moderately and positively with scores on the BEHS-6 measure of general health (higher self-rated health was associated with higher ERA-12-J scores), *r* = 0.30, *p* < 0.001. ERA-12-J total scores also correlated with self-rated health impact of the COVID-19 pandemic (higher impact associated with lower ERA-12-J score), *r* = −0.16, *p* < 0.001; and daily sitting minutes (lower sitting minutes associated with higher ERA-12-J score), *r* = −0.10, *p* < 0.001, although the correlation was relatively small in both instances. No significant associations were identified between ERA-12-J scores and education level, employment status or living situation.

## 4. Discussion

### 4.1. Summary of Main Findings

The construct of health-related expectations regarding aging is unique within gerontological research [12,16] and is potentially predictive of health-related behaviors and outcomes [8,15,23]. The aims of the study were to test the psychometric properties of the first Japanese translation of the ERA-12 scale and elucidate expectations across middle-aged and older adults. The ERA-12-J showed acceptable psychometric properties and conformed well to the three-subscale structure of the original English-language measure [16]. Specifically, the ERA-12-J showed acceptable test-retest reliability (*t*(1598) = 0.60, *p* = 0.63), internal consistency reliability (α > 0.80), inter-item correlation (*r* = 0.21–0.78), and item-total correlation (*r* = 0.53–0.73), and CFA verified the presence of the hypothesized three-factor structure on four common indices for fit (GFI, CFI, AGFI and RMSEA). These findings suggest that the ERA-12-J will provide an effective basis for the measurement of health-related expectations in Japan and useful data for international comparisons. Across the sample, the ERA-12-J revealed prevailing negative sentiments concerning physical, mental, and cognitive health. Significant and independent differences emerged on the basis of gender and age, with middle-aged adults and females holding more negative expectations about their future health. Notably, there were no interaction effects between age and gender in relation to health-related expectations about aging. This suggests that age and gender have an independent influence with regards to ERA-12-J total and subscales scores. The lack of an interaction effect may be explained by socio-cultural stratifications within Japanese society whereby gender roles and age-related distinctions between work and retirement ages potentially create powerful and discrete influences on health-related expectations [6,7]. Significant correlations were also identified between ERA-12-J and health-related variables, including self-reported impact of the COVID-19 pandemic, self-reported general health, and self-reported daily sitting behavior.

### 4.2. Alignment with International Studies

The psychometric properties of the Japanese version of the ERA-12 compared favorably to international versions of the measure. For instance, reported internal consistency reliability (α) for total scores of the English, Turkish and Singaporean versions were 0.89, 0.76 and 0.70 respectively (compared to 0.91–0.92 in the present study) [15,16,19]. Published international studies that have applied valid and reliable iterations of the ERA-12 have reported standardized scores in the following ranges: total score (26.42–46.33), physical (18.26–38.38), mental (41.52–61.10), and cognitive (19.30–40.08) [15,17,19,25]. The Japanese data for the ERA-12-J total and subscale scores fall comfortably within the ranges reported internationally and fit the pattern of lower scores on the physical and cognitive health subscales relative to mental health expectations. Negative overall and subscale sentiments about future health has previously been explained by high levels of individual awareness of, and attention towards, negative information about aging (e.g., media messages, agist stereotypes, or health information about vulnerability) [22,36]. Negative sentiments have potentially been reinforced during the current COVID-19 pandemic since the well-publicized risks of hospitalization and death increase significantly beyond age 60 [37]. Meta-analytic findings regarding the related concept of attitudes to aging have also shown comparatively negative views from collectivist cultures of north-east Asia (e.g., Japan, South Korea, and China) compared to more individualistic cultures [38]. Social norms in Japan may also contribute to negative expectations about health and capability in later life. For example, despite the world’s highest life expectancy and long-term reductions in the working age population, many industries in Japan maintain a mandatory retirement age regardless of individual intentions or capabilities [39]. Pervasive policies such as this potentially infer declining ability and health with age and reinforce outmoded beliefs about the necessity for older adult disengagement. 

While the Japanese ERA-12-J data conform closely with international findings, there are some unique results that require elucidation. Higher health-related expectations among male respondents was unexpected. This finding is counterintuitive when considering that Japanese females outlive males by a significant margin (>6 years), have comparatively better overall health, and engage in fewer risk behaviors [40,41]. Indeed, among international studies that have applied the ERA-12 it has been reported that women have significantly higher health-related expectations than men [26,36]. There are several explanations for this situation in Japan. Firstly, it may relate to the gendered nature of Japanese society whereby the relatively privileged position of males within the family, workplace and society may lead to higher (though unfounded) expectations about future life outcomes, including those related to physical, mental, and cognitive health [42,43,44]. It has been asserted that in a male dominated society there may be less acceptance of aging and age-related challenges among male populations even if health declines are more rapid in later life [42]. A second explanation may relate to a lack of knowledge about the long-term health risks of smoking and alcohol consumption or the normalization of self-medication with these substances linked to a lack of mental health support or prevalent work stress [45]. A third explanation for this outcome could be extreme heterogeneity within the Japanese male population. For example, it is well known that Japanese males (38%) have consistently high rates of tobacco use relative to females (13%) [44,46], yet males are also reportedly less likely to be physically inactive in national sport and activity surveys [47]. 

In addition to gender differences, significant variations were also identified in relation to age-cohort. Middle-aged respondents held significantly more negative health-related expectations relative to older adults. Observed differences between older and middle-aged cohorts contrast with some international findings whereby the oldest cohorts are often reported to hold more negative expectations [27,48]. However, Japanese data on heath behavior, mental health, and work environment support the present findings and provide a window into potential socio-cultural influences on health-related expectations. For example, national data collected on the physical activity behavior has shown that older adults often walk significantly more and sit less than middle-aged adults in Japan [6]. Other research has reported worse mental health outcomes (including suicide) among middle-aged people in Japan, which is surmised to relate to work and societal pressures [49]. The challenging nature of Japan’s work environment is well known internationally and characterized by demands for organizational loyalty, long working hours, expectations for weekend work and overtime, pervasive cultures of male authority, and infrequent use of annual leave [50,51]. Such conditions may limit discretionary time and associated activities, including participation in health promoting behaviors, and impair individuals’ perceptions of their future physical, mental, and cognitive health. 

### 4.3. Limitations and Strengths

There are limitations and strengths associated with the present research that researchers should be aware of when considering the results. Firstly, data were collected during the COVID-19 pandemic and may have been influenced by concerns about virus transmission or higher risks for serious illness among older adults [37]. It is possible that fears about the virus or individual vulnerability may have influenced self-reports regarding expectations for future health. Post-pandemic follow-up may be desirable to determine the level of influence, if any, from the conditions created by COVID-19. Secondly, the results are based on respondent self-reports that were submitted in an online survey. Such reports may be affected by social desirability biases or variations in internet accessibility. However, the large sample sizes (exceeding calculated power requirements) and high levels of internet usage in Japan (>90% of households) [52] may have mitigated these potential impacts. Additionally, it was notable that a large number of respondents over 75 years of age (*n* = 137) completed the online survey, which suggests that the format of administration did not discourage or limit responses among the oldest cohorts. To address these potential limitations, future research may include objective measures of current health state or behavior and both mail and online survey distribution to access potentially marginalized groups who may have limited internet connectivity (even if this population may ultimately be quite small). Strengths of this study include the use of large quota samples that exceed the respondent cohorts of most existing studies that have applied the ERA-12 and the use of CFA to verify the hypothesized factor structure. 

## 5. Conclusions

The ERA-12-J is a reliable and valid scale for measuring health-related expectations associated with aging in Japan. The results of the psychometric analysis and total/subscale scores conform closely to data published in the development of other versions of the scale, including English, Turkish, and Singaporean iterations. This suggests that the scale is likely to perform consistently in diverse societies. Considering the growing international use of the measure, with several validated and well-aligned translations now in use, the present study expands the potential for broader comparative cross-cultural studies of health-related expectations. The ERA-12-J has also highlighted important attributes of middle-aged and older adults in Japan by revealing gender and age-cohort challenges with regards to health-related expectations. Considering the potential associations between ERA-12 scores and health behaviors and outcomes that have been reported internationally, further investigation and intervention may be required in relation to middle-age and female cohorts in Japan. The path to improving health-related expectations for these groups (as well as general populations of middle-aged and older adults) is likely to require a multi-factorial approach that includes public health education (knowledge of risk factors and awareness of the positive benefits of lifestyle change), changes to work cultures (with a greater emphasis on work-life balance), and the continued empowerment of women in Japanese society.

## Figures and Tables

**Table 1 ijerph-19-13509-t001:** Demographic characteristics (N = 1600) and ERA-12-J scores.

Statistical Parameters		Time 1 (March 2021) *n* = 800	Time 2 (October 2021) *n* = 800
Age in years, Mean (SD)		62 (10.02)	62 (10.31)
Age range		45–93	45–94
Age group	45–64 years	400 (50%)	400 (50%)
	65 and older	400 (50%)	400 (50%)
Gender	Female	400 (50%)	400 (50%)
	Male	400 (50%)	400 (50%)
	Other/prefer not to say	0	0
Living situation	Live alone	186 (23.3%)	174 (21.8%)
	Live with family/spouse	608 (76.0%)	619 (77.4%)
	Other	6 (0.8%)	7 (0.9%)
Employment status	Full-time employee	250 (31.3%)	251 (31.4%)
	Part-time employee	126 (15.8%)	141 (17.6%)
	Home duties or retired	321 (40.1%)	286 (35.8%)
	Other	103 (12.9%)	122 (15.3%)
Highest education	Elementary school	1 (0.1%)	1 (0.1%)
	Middle school	11 (1.4%)	10 (1.3%)
	High school	173 (21.6%)	177 (22.1%)
	Vocational qualification	88 (11.0%)	80 (10.0%)
	Two-year college	95 (11.9%)	105 (13.1%)
	4-year degree	376 (47.0%)	374 (46.8%)
	Graduate school	48 (6.0%)	50 (6.3%)
	Other	8 (1.0%)	3 (0.4%)
Nationality	Japanese	794 (99.3%)	796 (99.5%)
	Other	6 (0.7%)	4 (0.5%)
ERA-12-J scores Mean (SD)/100	ERA-12-J Total score	36.80 (15.13)	37.16 (15.18)
	ERA-12-J physical (身体的)	28.41 (16.61)	28.22 (16.67)
	ERA-12-J mental (心的)	47.47 (19.98)	48.58 (20.09)
	ERA-12-J cognition (脳機能)	34.51 (17.08)	34.69 (18.03)

**Table 2 ijerph-19-13509-t002:** Internal consistency reliability.

Data Collection Period	Cronbach Alpha (α)
** *Time 1 (March 2021, n = 800)* **	
ERA-12-J total	0.92
ERA-12-J physical health subscale	0.91
ERA-12-J mental health subscale	0.85
ERA-12-J cognition subscale	0.90
** *Time 2 (October 2021, n = 800)* **	
ERA-12-J total	0.91
ERA-12-J physical health subscale	0.88
ERA-12-J mental health subscale	0.84
ERA-12-J cognition subscale	0.90

**Table 3 ijerph-19-13509-t003:** Results of CFA with middle-aged and older adult Japanese (N = 1600).

ERA-12-Japanese Item (and Original English Item)	Standardized Beta Coefficients		
	Factor 1 Physical Health	Factor 2 Mental Health	Factor 3 Cognitive Function
1. 私たちは加齢とともに，自分の健康状態が低下することを受け入れる必要がある。 *(When people get older, they need to lower their expectations of how healthy they can be.)*	0.791		
2. 私たちの身体は自動車のようなもので，古くなると様々な故障が起こる。 *The human body is like a car: When it gets old, it gets worn out*	0.869		
3. 身体のあちらこちらが痛むようになるのは，一種の老化現象であると思う。 *Having more aches and pains is an accepted part of aging*	0.813		
4. 加齢とともに，少しずつ活動力が低下してくると思う。 *Every year that people age, their energy levels go down a little more*	0.815		
5. 加齢とともに，友人や家族と過ごす時間が少なくなると思う。 *I expect that as I get older I will spend less time with friends and family*		0.609	
6. 加齢とともに，孤独さは増すものだと思う。 *Being lonely is just something that happens when you get old*		0.745	
7. 私たちは，加齢とともに不安なことが増える。 *As people get older they worry more*		0.780	
8. 加齢に伴い，気分が沈むのは普通のことである。 *It’s normal to be depressed when you are old*		0.824	
9. 加齢とともに，物忘れがひどくなるものだと思う。 *I expect that as I get older I will become more forgetful*			0.827
10. 加齢に伴い，物や人の名前を思い出せなくなるのは仕方ないことだ。 *It’s an accepted part of aging to have trouble remembering names*			0.845
11. 物忘れは，加齢に伴う自然な出来事である。 *Forgetfulness is a natural occurrence just from growing old*			0.895
12. 加齢とともに，頭の回転が遅くなることは避けられない。 *It is impossible to escape the mental slowness that happens with aging*			0.796
Estimated inter-correlation (latent variables), Factor 1 ->	-	0.531	0.696
Estimated inter-correlation (latent variables), Factor 2 ->	0.531	-	0.651
Estimated inter-correlation (latent variables), Factor 3 ->	0.696	0.651	-

**Table 4 ijerph-19-13509-t004:** Relationships between age, gender, and ERA-12-J subscale scores.

ERA-12 Subscales			Mean (SD)	*F*	*p*	η_p_^2^
Physical health subscale	Age group	Middle-aged	27.16 (18.53)	7.81	0.005	0.01
		Older	29.47 (14.41)			
	Gender	Females	26.96 (16.91)	10.71	0.001	0.01
		Males	29.67 (16.25)			
	Age*gender			0.006	0.94	0.00
Mental health subscale	Age group	Middle-aged	46.06 (21.42)	15.56	<0.001	0.01
		Older	49.99 (18.35)			
	Gender	Females	46.69 (20.54)	7.23	0.007	0.01
		Males	49.34 (19.43)			
	Age*gender			0.12	0.73	0.00
Cognitive health subscale	Age group	Middle-aged	33.88 (19.45)	2.75	0.10	0.00
		Older	35.32 (15.41)			
	Gender	Females	32.99 (17.31)	13.57	<0.001	0.01
		Males	36.21 (17.66)			
	Age*gender			1.18	0.28	0.00

## Data Availability

Data upon which this research is based are available from the lead author upon reasonable academic request and at the discretion of the authorship team.

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
