# Peer review of "Health-Related Expectations Regarding Aging among Middle-Aged and Older Japanese: Psychometric Performance and Novel Findings from the ERA-12-J"

_ijerph, 2022, doi:10.3390/ijerph192013509_

Round 1

Reviewer 1 Report

Thank you for your effort for this paper. This study is meaningful in that it evaluated the measurement properties of the ERA-12 tool to measure health-related expectations. However, the following parts need to be supplemented.

-      Describe the respondent's characteristics in detail. What the characteristics of the population living in large city (Tokyo) and using e-commerce platforms, i.e. what was the composition of age and gender. And what methods were used to allocate them, etc. This is because the characteristics and sampling methods of the target population can cause systematic errors and affect the internal validity of the results. Next, when discussing comparisons with international research results, discuss what these characteristics mean to the results.

-      For consistency of analysis methods and result, describe in Analysis section that the Cronbach's alpha coefficient to evaluate the internal consistency reliability of the ERA-12-J.

-      Test-retest reliability is a specific way to measure reliability of a test and it refers to the extent that a test produces similar results over time. Test-retest reliability is a measure of reliability obtained by administering the same test twice over a period of time to a group of individualsHowever, in your study, the subjects at time 1 and time 2 are not the same respondents. Please provide the rationale for using this method.

-      Two-way ANOVA focuses on interactions between independent variables rather than on the main effects of individual independent variables of interest. However, your research tends to emphasize the main effect. Discuss the non-significant interaction effects.

Author Response

We offer our sincere thanks for your constructive and timely feedback.

Please see the attached cover letter with full point-by-point responses to your comments.  

Reviewer 2 Report

First of all, I would like to mention that it has been a pleasure for me to review this article. For this reason, I would like to convey my sincere thanks to the editorial team of the journal and the authors.

The article deals with a highly relevant topic as it has given the Japanese population a valid and reliable tool to know the expectations of the elderly. However, I would like to make the following suggestions to the authors:

-       - The results subsection of the abstract shows that the ERA-12 scale has acceptable test-retest reliability, internal consistency, inter-item reliability and item-total reliability. However, it would be advisable to specify the results with statistical data. In fact, the information presented in the results would be a clarification to be added in the discussion for comparison with other versions of the scale.

-       - In the translation process, was a back-translation of the Japanese version carried out to check the similarity of content with the original version? This methodological aspect may be relevant for the results.

-      -  What criteria were used to test in March and restest in October, and how did this time frame interfere with the results? Did the same subjects complete the scale at both times?

-      - In the section on limitations of the study, the authors refer to those related to data collection during the pandemic, because of their possible influence on health expectations/concerns. Do you think that the format of data collection could be a limitation?

Author Response

(The authors gave the same response as above.)

Round 2

Reviewer 1 Report

I suggest that some minor revisions are needed.

- In 'Abstract' and '4.1. Summary of main findings'

Present 'inter-item reliability' and 'item-total reliability' on a minimum to maximum range. And record the correlation coefficient, r.

- Please correct the title in Table 2:

Table 2. International consistency reliability -> Internal consistency reliability

Author Response

We thank the reviewers for their helpful comments and requests for further minor revisions. Please see our point by point responses below and updates in the manuscript. 

Reviewer comment: In 'Abstract' and '4.1. Summary of main findings' Present 'inter-item reliability' and 'item-total reliability' on a minimum to maximum range. And record the correlation coefficient, r.

Author response: We have now added the range of values (min-max) for inter-item and item-total correlations as requested in the abstract (page 1, line 23) and in the discussion (summary of main findings, page 9, line 262). We have also added the 'r' correlation coefficient label as requested in both locations.  

Reviewer comment 2: Please correct the title in Table 2: Table 2. International consistency reliability -> Internal consistency reliability.

Author response: The heading of Table 2 has been updated (see page 6). Thanks for this correction.